# Improving Arctic Surface Radiation Estimation Using a Nonlinear Perturbation Model with a Fused Multi-Satellite Cloud Fraction Dataset

Yueming Zheng<sup>1</sup>, Tao He<sup>1</sup>, Yichuan Ma<sup>2</sup>, Xinyan Liu<sup>3</sup>

- <sup>1</sup>School of Remote Sensing and Information Engineering, Wuhan University, Hubei, 430070, China
  <sup>2</sup>Jockey Club STEM Laboratory of Quantitative Remote Sensing, Department of Geography, The University of Hong Kong, Hong Kong, 999077, China
  - <sup>3</sup>Aerospace Information Research Institute, Henan Academy of Science, Henan, 450100, China
- 10 Correspondence to: Tao He (taohers@whu.edu.cn)

Abstract: Arctic has been undergoing rapid climate change, where radiative processes are key controlling factors. However, cloud-related uncertainties remain the primary barrier to accurately estimating the radiation budget. Strong coupling between clouds and other variables complicates the isolation of cloud-related impacts, and the linear assumptions in traditional models further restrict attribution of radiation changes to cloud-related influences. This study introduces an artificial neural network model that emulates radiative components typically represented in radiative transfer or climate models. Without relying on linear assumptions, the model directly quantifies the influence of cloud fraction (CF) on radiation. Using a more accurate CF dataset, we refined the monthly downwelling shortwave radiation (DSR) estimates from Clouds and the Earth's Radiant Energy System (CERES) SYN products and further estimated all-wave net radiation (NR) from the corrected DSR. Validation against ground-based observations confirmed that the CF-corrected DSR effectively mitigated the overestimation in CERES DSR, reducing biases by up to 23 W m<sup>-2</sup>. At sites where CF underestimation exceeded 25%, the monthly-mean bias decreased from 25.70 W/m<sup>2</sup> to 4.88 W/m<sup>2</sup>, with RMSE reduced from 40.36 W/m<sup>2</sup> to 32.60 W/m<sup>2</sup>. The estimated monthly NR also improved markedly (RMSE reduced from 34.88 W/m<sup>2</sup> to 28.90 W/m<sup>2</sup>). Under large CF underestimation (>30%), the CERES NR nearly failed (R<sup>2</sup> = 0.0182), whereas NR derived from CF-corrected DSR retained reasonable agreement (R<sup>2</sup> = 0.5411). Importantly, this work produces a new NR dataset with enhanced accuracy over the Arctic, offering direct value for studies of surface energy balance, climate feedbacks, and long-term variability.

## 1 Introduction

The Arctic is currently warming at rates two to four times higher than those in lower latitudes (Cohen et al., 2020; Coulbury and Tan, 2024; Rantanen et al., 2022), a phenomenon known as Arctic amplification (Previdi et al., 2021). This rapid warming profoundly alters the Arctic climate system, affecting sea ice, permafrost, and atmospheric circulation patterns (Bennartz et al., 2013; Goosse et al., 2018). Among the key physical quantities that govern energy exchange in the climate

system, surface net radiation (NR) —the net sum of between incoming and outgoing shortwave and longwave radiation at the surface—plays a central role in determining surface energy budgets, ice—albedo feedback, and surface temperature variability (Brown and Caldeira, 2017; Goessling et al., 2025; Huang et al., 2021; Loeb et al., 2022).

Despite its importance, current estimates of radiation over polar regions remain highly uncertain. According to the Intergovernmental Panel On Climate Change (IPCC, 2023), satellite-based assessments indicate that the uncertainties in global monthly mean fluxes are about 10 W m<sup>-2</sup>, while those over polar regions are even larger than other regions. These uncertainties can distort the assessment of surface energy balance, compromise the accuracy of global temperature trend estimates, and reduce the simulation reliability of changes in cryosphere (Cheng et al., 2017; Loeb et al., 2021; Prince and L'Ecuyer, 2024). Lee et al. (2019) showed that discrepancies in surface radiative fluxes among reanalysis datasets (up to ~60 W m<sup>-2</sup> in shortwave) can substantially affect Arctic sea-ice simulations, such flux differences led to sea-ice volume changes of 3000–4000 km<sup>3</sup> and altered interannual variability by up to 40%. This highlights the strong sensitivity of Arctic sea ice to uncertainties in surface radiation.

45 Improving the radiation estimates in the Arctic is therefore an urgent task, especially in the context of detecting early signals of anthropogenic influence and validating climate model projections (Danso et al., 2020; Van Tricht et al., 2016; Zheng et al., 2025). However, achieving this goal remains challenging due to the complex radiative interactions between surface and atmosphere. Among them, clouds represent a dominant source of uncertainty (Tao et al., 2025). Clouds regulate the Earth's radiation balance by reflecting incoming shortwave radiation and absorbing/emitting longwave radiation (Dessler, 2010; McCoy et al., 2017; Sledd and L'Ecuyer, 2019). However, in the Arctic, accurately detecting cloud remains challenging, particularly for passive satellite sensors. Clouds are characterized by numerous parameters such as optical depth, phase, height, and particle size, retrieving them consistently from spaceborne measurements is inherently difficult. Among these variables, cloud fraction (CF) is arguably a simplified descriptor, yet it has become the most fundamental parameter for radiation studies owing to its availability across multiple satellite products with long temporal coverage and broad spatial extent. For instance, Liu (2022) showed that limitations in detecting low-level Arctic clouds with active sensors (25 % underestimation) can introduce errors in monthly mean cloud radiative forcing of up to ~2.5 W m<sup>-2</sup> at the surface and ~3.4 W m<sup>-2</sup> at the TOA. Zib et al. (2012) evaluated five reanalysis datasets against BSRN observations at Arctic sites and found that CF biases of up to 20-30% could lead to monthly deviations in downward shortwave radiation (DSR) exceeding 90 W m<sup>-2</sup>. Wei et al. (2021), analyzing CMIP6 models showed that systematic CF overestimation of about 5-15% in the Arctic resulted in positive biases in NR greater than 3 W m<sup>-2</sup>. These findings suggest that CF is a substantial contributor of uncertainty in Arctic radiation estimation. Therefore, quantitatively constraining the radiation biases attributable to CF errors could enable the direct use of more accurate CF datasets to reduce uncertainties in radiation products.

Building on this motivation, many perturbation-based methods have been developed to quantify the radiation impacts of CF (Table 1). For instance, the partial radiative perturbation (PRP) method can isolate the contribution of individual variables but requires repeated radiative transfer calculations and often yields inconsistent feedback estimates across models (Colman, 2003; Soden et al., 2004; Wetherald and Manabe, 1988). The radiative kernel technique improves efficiency by precomputing flux sensitivities, yet its assumption of linearity breaks down when cloud–radiation interactions are strongly nonlinear, leading to significant errors (Shell et al., 2008; Soden et al., 2008; Zhu et al., 2019). Moreover, the structure of the kernel itself varies across models, introducing model dependence and affecting cloud feedback estimates (Hahn et al., 2021; Jonko et al., 2012). Even satellite-based variants such as CERES-PRP reduce some model dependence but still demand multiple computations and remain limited by the accuracy of the underlying CF datasets (Thorsen et al., 2018).

To address these challenges, we developed a nonlinear perturbation method that captures the relationship between CF perturbations and shortwave radiation changes under various atmospheric and surface conditions. Another major advancement of this study lies in the coupling of this method with a more accurate CF dataset that we previously developed (Liu et al., 2023). Unlike standard passive-sensor cloud products, this dataset employs spatiotemporal fusion techniques to merge CF information from active and passive satellite observations as well as ground-based measurements. Its fused, multi-source design substantially reduces biases in CF estimates and ensures improved spatiotemporal completeness. By coupling the nonlinear perturbation method with this enhanced CF dataset, we effectively corrected CF-induced DSR biases in the Clouds and the Earth's Radiant Energy System (CERES) SYN. Furthermore, to minimize potential error propagation arising from independently adjusting shortwave and longwave components, we extended the method to directly estimate the NR based on the corrected DSR. The objectives of this study are: (i) to construct a nonlinear perturbation relationship between CF and DSR based on satellite observations, and apply it to correct Arctic DSR with improved CF products; (ii) to develop an extended model for directly estimating NR based on the corrected DSR.

Table 1. Summary of perturbation-based methods for quantifying radiative impacts of CF and other climate variables (the current paper is added for completeness).

| Reference                          | Methods                                                                                                                                               | Data                                          | Assumptions                                                                                                                                | Key Results                                                                                                                                                   |
|------------------------------------|-------------------------------------------------------------------------------------------------------------------------------------------------------|-----------------------------------------------|--------------------------------------------------------------------------------------------------------------------------------------------|---------------------------------------------------------------------------------------------------------------------------------------------------------------|
| (Wetherald<br>and Manabe,<br>1988) | Partial Radiative Perturbation (PRP): replace one variable (e.g., CF) in a control climate and compute flux difference while holding others constant. | Early GCM simulations.                        | Feedback can be isolated<br>by substituting a single<br>variable while all others<br>remain unchanged; linear<br>superposition of effects. | First method to separate individual feedbacks; computationally very demanding; can yield inconsistent estimates depending on model setup.                     |
| (Colman, 2003)                     | Inter-model<br>comparison of<br>feedbacks in GCMs<br>using PRP and other<br>approaches.                                                               | Multiple GCM experiments.                     | Feedbacks are<br>comparable across<br>models if methods are<br>applied consistently.                                                       | Found substantial spread<br>among models in feedback<br>strengths, pointing to<br>methodological and structural<br>uncertainties.                             |
| (Shell et al., 2008)               | Radiative Kernel (RK)<br>technique: pre-compute<br>kernels from NCAR<br>CAM to quantify<br>sensitivity of fluxes to<br>small perturbations.           | NCAR<br>Community<br>Atmospheric<br>Model.    | Radiative response is<br>linear for small<br>perturbations; kernels<br>can be applied to other<br>simulations.                             | Provided an efficient<br>framework to estimate<br>feedbacks; avoided repeated<br>full radiative transfer<br>calculations; limited by<br>linearity assumption. |
| (Soden and<br>Held, 2006)          | Assessment of RK-based feedbacks in coupled ocean—atmosphere GCMs.                                                                                    | Coupled model simulations.                    | Small perturbations yield linear flux–variable relationship.                                                                               | Demonstrated usefulness of<br>RK for large-scale feedback<br>analysis, but highlighted<br>model dependence of kernels.                                        |
| (Soden et al., 2008)               | Standardization of RK framework across models.                                                                                                        | Multi-model ensembles.                        | Linearity of flux responses across different forcings.                                                                                     | Enabled cross-model intercomparison of feedbacks; showed kernels improve efficiency, but neglect nonlinear interactions.                                      |
| (Zhu et al.,<br>2019)              | Neural network (NN) approach to estimate climate feedbacks.                                                                                           | GCM outputs used for training and validation. | NN can learn nonlinear relationships without explicit assumptions.                                                                         | Demonstrated that NN reduces<br>errors from nonlinear<br>interactions, improving<br>accuracy over RK/PRP.                                                     |
| (Hahn et al., 2021)                | Polar amplification analysis with RK-based methods.                                                                                                   | CMIP5 and<br>CMIP6 model<br>outputs.          | Choice of kernel sufficiently represents model physics.                                                                                    | Found that cloud feedback estimates are highly sensitive to kernel choice; kernel dependence introduces additional uncertainty.                               |
| (Jonko et al.,                     | Extended RK approach under changing CO <sub>2</sub>                                                                                                   | CCSM3 climate model                           | Linear kernel assumption valid across a wide range                                                                                         | Showed that when perturbations are large, linear                                                                                                              |

| 2012)                  | forcing.                                                                                                                           | simulations.                                                                     | of forcings.                                                                                                                   | kernels break down, limiting reliability.                                                                                                                                                                                                            |
|------------------------|------------------------------------------------------------------------------------------------------------------------------------|----------------------------------------------------------------------------------|--------------------------------------------------------------------------------------------------------------------------------|------------------------------------------------------------------------------------------------------------------------------------------------------------------------------------------------------------------------------------------------------|
| (Thorsen et al., 2018) | CERES-PRP method:<br>observation-based<br>extension of PRP,<br>using satellite<br>radiances to<br>decompose flux<br>perturbations. | CERES satellite products.                                                        | Observed flux perturbations can approximate single-variable contributions; nonlinear effects handled by repeated calculations. | Provided first observation-<br>based PRP estimates; allowed<br>flexible variable<br>combinations; still<br>computationally intensive and<br>not free of nonlinearity.                                                                                |
| This study             | Nonlinear perturbation<br>method; extended to<br>estimate all-wave<br>surface net radiation                                        | Multi-source<br>fused CF<br>dataset,<br>satellite and<br>ground<br>measurements. | Nonlinear CF–radiation<br>relationships can be<br>empirically captured<br>from observations                                    | First to apply CF perturbation correction with ANN in the Arctic; effectively reduced DSR overestimation at high CF biases. Developed extended model linking corrected DSR to NR; highlighted that CF accuracy critically determines NR performance. |

#### 2 Data and materials

# 2.1 CERES SYN1deg




The CERES SYN1deg product (Level 3) provides monthly, daily, and hourly averages of TOA fluxes based on the Angular Distribution Model, as well as TOA and surface fluxes derived from a radiative transfer model (Rutan et al., 2015). Validation against Baseline Surface Radiation Network (BSRN) observations shows that hourly all-sky shortwave fluxes errors are typically within ±1% globally and about -1.9% poleward of 60°, with RMSE around 12.6 W m<sup>-2</sup> globally and 19.4 W m<sup>-2</sup> in the Arctic; monthly RMSE values are 5.7 W m<sup>-2</sup> globally and 20.2 W m<sup>-2</sup> at high latitudes (NASA POWER Project, 2024). The SYN1deg Ed4A product integrates fluxes observed by CERES with cloud properties derived from Moderate Resolution Imaging Spectroradiometer (MODIS) and Geostationary (GEO) image products(Doelling et al., 2013, 2016). Cloud retrievals are based on the CERES Cloud Working Group Ed4A algorithms (Minnis et al., 2011, 2021; Trepte et al., 2019), and above 60° latitude the SYN cloud products are fully consistent with those from SSF. Given that the SYN product provides only diurnal monthly averages of cloud properties, and that shortwave radiation is exclusively influenced by daytime cloud, this study utilizes cloud products sourced from the SSF, which offers daytime cloud data.

## 2.2 Fused Cloud Fraction of Arctic

Fused\_cf\_Arc is a daytime Arctic CF product developed by Liu et al. (2023) that employs spatiotemporal fusion methods, utilizing cumulative distribution function matching and Bayesian maximum entropy to integrate CF products from active satellites, passive satellites, and ground observations. This product spans the period from 2000 to 2020, offering a seamless monthly daytime average for the Arctic region with a spatial resolution of 1 degree. Compared to other products, Fused\_cf\_Arc provides more comprehensive spatiotemporal coverage and enhanced accuracy. Its coefficient of determination (R²) improves by approximately 0.20 to 0.48 when compared to products such as MOD08, CERES-SSF, International Satellite Cloud Climatology Project (ISCCP), Cloud, Albedo, and Radiative Transfer Algorithm (CLARA), and Cloud-Aerosol Lidar and Infrared Pathfinder Satellite Observations (CALIPSO). In terrestrial areas, the root mean square error (RMSE) and bias are reduced by approximately 6.09% and 4.04%, respectively. In marine regions, these metrics improve by 0.05 to 0.31, 2.85%, and 3.15%. Notably, the performance of Fused\_cf\_Arc is most significantly enhanced in the Greenland.

# 2.3 Merra-2 Reanalysis Data

MERRA-2 is a suite of global atmospheric reanalysis products developed by the NASA Global Modeling and Assimilation Office, incorporating recent advancements in modeling and data assimilation (Gelaro et al., 2017). It provides monthly global air temperature (Ta) and specific humidity data from 1980 onward. For the estimation of NR, the extended model





required Ta and relative humidity (RH) as input variables. While RH is not directly available in MERRA-2, it can be derived from its specific humidity data.

## 120 2.4 GLASS FVC Product

Surface vegetation affects the relationship between shortwave radiation and NR (Chen et al., 2022). In this study, the GLASS fractional vegetation cover (FVC) was selected for its comprehensive spatiotemporal coverage and high accuracy. Jia et al. (2015) developed a robust FVC retrieval algorithm based on MODIS surface reflectance data, achieving an RMSE of 0.157 when validated against ground-based measurements. After applying gap-filling techniques, the GLASS FVC became spatially continuous at the global scale, making it highly suitable for various vegetation monitoring and research applications (Wang et al., 2020). This dataset features a spatial resolution of 500 m and a temporal resolution of 8 days.

#### 2.5 Ground-measured data

Ground measurements provide accurate surface radiation observation data that can be used to validate our results. In this study, ground measurement data were obtained from four reliable radiation flux observation networks: FLUXNET, AmeriFlux, Global Energy Balance Archive (GEBA), and The Programme for Monitoring of the Greenland Ice Sheet (PROMICE) (Fig.1).

FLUXNET is one of the largest global flux observation networks, integrating over 800 active and historical sites that span most climate zones and representative biomes to monitor carbon, water, and energy fluxes in terrestrial ecosystems. For this research, 13 sites that include data on radiation were selected for analysis. Within this global network, AmeriFlux serves as the North American regional component, supported by the U.S. Department of Energy and partner agencies. Established in 1996 with about 15 sites, AmeriFlux has expanded to more than 110 active towers across diverse ecosystems. For this study, we used data from 18 AmeriFlux sites located north of 60°N that include radiation measurements, primarily in northern and western Alaska.

The GEBA database is designed to centrally store measurements of surface energy flux from around the world, maintained by the Swiss Federal Institute of Technology Zurich. This database includes observational data for 15 components of surface energy flux, strictly comprising only directly measured surface fluxes and excluding empirically derived fluxes. Gilgen et al. (1998) provide a detailed description of the error estimates and quality checks applied to these data. The Arctic contains a significant number of GEBA sites, including ocean buoys and ground observation stations. However, most of these sites have relatively short operational periods. In this study, 22 sites within the 2000-2020 were selected for validation.

The PROMICE is a project aimed at monitoring changes in the Greenland ice sheet, operated by Denmark, Greenland, and other partners. PROMICE has established a network across the western, central, and eastern regions of Greenland to monitor

variables such as surface elevation changes, snow depth, temperature, humidity, and the impact of global climate change on the ice sheet. In this study, 14 sites from the PROMICE network, covering the period from 2000 to 2020, were selected as validation data.

Figure 1: Spatial distribution of 66 ground stations from four radiation flux networks. The base map was designed and developed by Esri. For more information on this map, visit https://goto.arcgisonline.com/maps/Arctic\_Ocean\_Base.

## 3 Methodology


The improved estimation of NR was carried out in two steps. First, we developed a nonlinear perturbation model based entirely on satellite observations, using an artificial neural network (ANN) to quantify DSR biases induced by CF errors. This model was applied to correct the DSR from CERES SYN using a more accurate CF dataset. In the second step, to avoid the propagation of errors caused by separately correcting shortwave and longwave components, we incorporated additional auxiliary variables—including MERRA-2 air temperature (Ta) and relative humidity (RH), as well as GLASS fractional vegetation cover (FVC)—and constructed an extended model to directly estimate NR based on the corrected DSR.

# 3.1 Bias Correction of Shortwave Radiation Using a Nonlinear CF Perturbation Model

#### 165 3.1.1 Theoretical framework

The theoretical foundation of the methods used in this study is based on the observation-based radiation kernel approach (CERES-PRP) proposed by Thorsen et al. (2018), which has been further improved. CERES-PRP employs the NASA Langley Fu-Liou radiation transfer model, with model inputs primarily derived from various observational datasets (monthly means) used in the processing of CERES version 4 data products. CERES-PRP calculates the radiation perturbation  $\delta F_{\Delta x,CM}$  caused by some perturbation  $\Delta x$  of variable x using the central difference method:

$$\delta F_{\Delta x,CM} = \frac{\delta F_{\Delta x,C}^f + \delta F_{\Delta x,M}^b}{2} + O_{CM}(\Delta x^2), \qquad (1)$$

$$\delta F_{\Delta x,C}^f = F(\overline{x} + \Delta x, \overline{y}_1, ..., \overline{y}_n) - F(\overline{x}, \overline{y}_1, ..., \overline{y}_n) + O_C^f(\Delta x), \qquad (2)$$

$$\delta F_{\Delta x,M}^{b} = F(x, y_{1}, ..., y_{n}) - F(x - \Delta x, y_{1}, ..., y_{n}) + O_{M}^{b}(\Delta x),$$
(3)

where the fluxes F are at some level of atmospheric level (TOA or surface), calculated using NASA Langley Fu-Liou radiation transfer model.  $\delta F_{\Delta x}^f$  and  $\delta F_{\Delta x}^b$  represent the effect of perturbation  $\Delta x$  calculated using forward and backward finite differences, respectively. The subscript C and M denote that the flux perturbation is relative to the climatological monthly mean base state and the monthly mean base state, respectively. x denotes the monthly mean, and  $\Delta x$  is the deseasonalized anomaly in the monthly mean value x relative to the climatological monthly mean  $\bar{x}$ , defined as  $\Delta x = x - \bar{x}$ .  $(y_1, ..., y_n)$  refers to the monthly means of all other variables related to radiation transfer.  $(\bar{y}_1, ..., \bar{y}_n)$  denote the climatological monthly means.  $O_{CM}(\Delta x^2)$  represents the minimized truncation error.

Since the fluxes computed relative to the monthly  $(y_1, ..., y_n)$  and climatological  $(\bar{y}_1, ..., \bar{y}_n)$  have different time average-related biases, this reduces the accuracy of Eq.1. The  $\delta F_{\Delta x, CM}$  can also be calculated using the following formula:

$$\delta F_{\Delta x,C}^{b} = F(\overline{x}, \overline{y}_{1}, ..., \overline{y}_{n}) - F(\overline{x} - \Delta x, \overline{y}_{1}, ..., \overline{y}_{n}) + O_{C}^{b}(\Delta x), \qquad (4)$$

185 
$$\delta F_{\Delta x,M}^f = F(x + \Delta x, y_1, ..., y_n) - F(x, y_1, ..., y_n) + O_M^f(\Delta x),$$
 (5)

First, the radiation perturbation  $\delta F_{\Delta x}$  is calculated using the monthly mean-based equations (Eq.3 and Eq.5). If, during the calculation, the variable  $x + \Delta x$  results in non-physical values (e.g., for CF perturbations,  $x + \Delta x$  may become less than 0 or greater than 100), the rules outlined by Thorsen et al. (2018) are followed. In this case, either Eq.2 or Eq.4 is selected for the calculation, or the cloud boundary perturbation is adjusted to be as close as possible to the true perturbation without being unphysical. Finally, all valid calculation results are averaged to obtain the final central differencing  $\delta F_{\Delta x}$ . Since the calculation of  $\delta F_{\Delta x}$  requires intensive radiative transfer computations, to enhance computational efficiency, the RK is precalculated. Multiple calculations are performed as needed to avoid nonlinearity in the RK. In subsequent applications, the  $\delta F_{\Delta x}$  can be obtained directly by multiplying the known perturbation  $\Delta x$  by the corresponding kernel coefficient  $K_{\Delta x}$ :

$$K_{\Delta x} = \frac{\delta F_{\Delta x}}{\Delta x} \tag{6}$$

# 195 3.1.2 Nonlinear CF Perturbation Model

190

To replace the extensive radiation transfer calculations (Eq.1 to 5) and further avoid additional pre-computation of the RK (Eq.6), we employ an ANN model to construct the relationship between the perturbation variable (CF) and shortwave fluxes. Subsequently, the radiative perturbation  $\delta F_{\Delta cf}$  induced by a change  $\Delta cf$  in the perturbation variable CF can be directly

derived from Eq.7. The corrected radiation estimate, obtained using more accurate CF products, can be derived by adding the perturbation radiation caused by the CF difference to the original radiation values.

One significant advantage of the ANN is its ability to learn from data without explicit rules, automatically extracting features and patterns (Zheng et al., 2024). We used the Backpropagation (BP) algorithm to train the model, with 40 neurons in the hidden layer, a maximum of 400 iterations, a learning rate of 0.03, and a target error of 0.00004, while other parameters were set to their default values. The model inputs were derived entirely from the monthly means of the CERES version 4 dataset, including cloud properties and other meteorological variables related to radiation transfer theory (Table 2), with the dependent variables being shortwave radiation, covering data from April to September 2003 to April to September 2020. 70% of the data was randomly selected for training, while 30% was used for validation.

$$\delta F_{\Delta cf} = \frac{f_{ANN}(cf + \Delta cf, y_1...y_n) - f_{ANN}(cf - \Delta cf, y_1...y_n)}{2} + O(\Delta cf^2)$$

$$(7)$$

where  $f_{ANN}$  represents the nonlinear function linking input variables and shortwave radiation,  $F_{SYN}$  is the radiation from the CERES SYN, cf is from the CERES SSF CF,  $\Delta cf$  represents the difference between the CF in the CERES SSF and the Fused\_cf\_Arc, non-physical perturbation values are addressed following the rules proposed by (Thorsen et al., 2018), and  $O(\Delta cf^2)$  is the truncation error in the ANN model calculation process. Given the ANN model's strong capability in handling nonlinear relationships, the experimental results for this term are nearly zero, and therefore, it is neglected. Finally, the shortwave radiation corrected by the higher-accuracy CF can be expressed as follows:

$$F_{corrected} = F_{SYN} + \Delta F_{\Delta cf} \tag{8}$$

Table 2: Input variables used in ANN model.

|                | Source                                                                |                          |  |
|----------------|-----------------------------------------------------------------------|--------------------------|--|
| $\Delta x$     | Base-states conditions                                                | -                        |  |
| Cloud fraction | Cloud top pressure, Cloud base pressure, phase, Cloud visible optical | CERES SSF, Monthly means |  |
|                | depth, Cloud ice/water particle radius, Cloud liquid/ice water path,  |                          |  |
|                | Cloud emissivity.                                                     |                          |  |
|                | Aerosol optical depth, Column Ozone, Skin temperature, Relative       | CERES SYN, Monthly means |  |
|                | Humidity, Surface albedo, Solar insolation.                           |                          |  |

## 220 3.2 Extended Model for all-wave net radiation Estimation

Shortwave radiation budget plays a crucial role in the formation of NR. Building upon our previous research (Chen et al., 2022), we further leveraged the strong correlation between shortwave radiation and NR to develop an extended estimation

225

model based on shortwave radiation. Previous studies have demonstrated that vegetation and temperature are key factors in regulating the relationship between shortwave radiation and NR (Huang et al., 2016; Jiang et al., 2015). Additionally, the length of daylight (LRD) has been found to influence their daily mean relationship (Chen et al., 2020). Therefore, in constructing the extended model, we incorporated shortwave radiation, surface albedo, LRD, fractional vegetation cover (FVC), Ta, and RH as key variables, all aggregated to monthly means as model inputs. Ultimately, the NR was estimated as follows:

$$NR = f(DSR, albedo, LRD, FVC, T_a, RH)$$
 (9)

where *f* is constructed using an ANN model, NR, DSR, and albedo are derived from CERES observations. The LRD is calculated pixel by pixel based on solar angles and pixel latitude and longitude. FVC is obtained from the GLASS product, *Ta* and *RH* are sourced from MERRA-2 or derived from its data.

#### 3.3 Evaluation statistics

We conducted a comparative evaluation of the radiation before and after correction using ground station data, employing the following three statistical metrics:

$$R^{2} = \frac{\left(\sum (y_{i} - \overline{y})(Y_{ground,i} - \overline{Y_{ground}})\right)^{2}}{\left(\sum (y_{i} - \overline{y})^{2}\right)\left(\sum (Y_{ground,i} - \overline{Y_{ground}})^{2}\right)},\tag{10}$$

$$RMSE = \sqrt{\frac{\sum (y_i - Y_{ground,i})^2}{N}},$$
(11)

$$Bias = \frac{\sum (y_i - Y_{ground,i})}{N},$$
(12)

where  $y_i$  is the radiation before/after correction,  $\bar{y}$  is the mean of the radiation before/after correction.  $Y_{ground,i}$  is the ground-based measured radiation, and  $\overline{Y_{ground}}$  is the mean of the ground-based measured radiation.

#### 4 Results

## 4.1 CF Perturbation Model training and validation

As there are obvious seasonal variations in radiation, we developed separate ANN models for each month. Table 3 presents the accuracy of all models on the training and validation datasets. All models exhibit an R<sup>2</sup> value greater than 0.98 on both the training and validation datasets, with most reaching 0.99. The root-mean square errors (RMSEs) are below 7.7 W/m<sup>2</sup>, showing slight variation across months, peaking in June and reaching its lowest in September. For all models, the RMSEs during the validation process is slightly lower than that during the training process, indicating that the models did not overfit. Overall, the results demonstrate that our models possess high reliability.

Table 3: Training and validation accuracy of CF-Radiation Perturbation Model. Note 'Month\_Apr\_CF-DSR' refers to the CF-DSR perturbation model for April.

| Model            | Training       |                         | Validation     |                         |
|------------------|----------------|-------------------------|----------------|-------------------------|
| _                | $\mathbb{R}^2$ | RMSE(W/m <sup>2</sup> ) | $\mathbb{R}^2$ | RMSE(W/m <sup>2</sup> ) |
| Month_Apr_CF-DSR | 0.98           | 6.38                    | 0.98           | 6.34                    |
| Month May CF-DSR | 0.98           | 7.73                    | 0.98           | 7.70                    |
| Month Jun CF-DSR | 0.99           | 7.17                    | 0.99           | 7.08                    |
| Month Jul CF-DSR | 0.99           | 6.04                    | 0.99           | 5.99                    |
| Month Aug CF-DSR | 0.99           | 3.88                    | 0.99           | 3.72                    |
| Month_Sep_CF-DSR | 0.99           | 2.19                    | 0.99           | 2.15                    |

## 4.2 Validation of corrected DSR against ground measurements

Figure 2 presents the validation of our corrected DSR estimates using all ground station data, alongside the CERES SYN DSR for comparison. The results show improvements across all metrics following adjustment. Specifically, R<sup>2</sup> increased from 0.8453 to 0.8527, RMSE decreased from 28.14 to 27.04 W/m<sup>2</sup>, and bias reduced from 1.18 to -0.06 W/m<sup>2</sup>. The corrected DSR align more closely with the 1:1 line, particularly showing notable improvements in addressing the overestimation of DSR. These results are expected, as our cloud product provides more comprehensive coverage, capturing higher daytime CFs compared to CERES (Liu et al., 2023), which leads to reduced DSR values.

We further present the validation results for sites where CERES significantly underestimates CF relative to Fused\_cf\_Arc (Fig. 3), which more clearly highlight the performance of the corrected DSR. For sites with CF underestimation exceeding 10%, RMSE decreased from 33.09 to 30.37 W/m², and bias reduced from 10.98 to 1.01 W/m². For sites with CF underestimation greater than 20%, RMSE decreased from 38.36 to 32.37 W/m², with a remarkable bias improvement, dropping by nearly 20 W/m² (from 22.41 to 3.68 W/m²). These results demonstrate that the DSR corrected using the more accurate cloud product shows substantial improvements under conditions where CERES CF is severely underestimated.

Figure 2: Validation of DSR against all ground stations before and after correction.

280

285

Figure 3: Validation of DSR against ground stations before and after correction at sites where CERES CF is underestimated relative to Fused\_cf\_Arc by more than 10% (a) and 20% (b). Blue represents the values before correction, while red represents the values after correction.

We further examined the DSR at stations where CERES CF was increasingly underestimated relative to Fused\_cf\_Arc, in order to evaluate the performance of the correction under different levels of CF underestimation. The results reveal that both RMSE and bias are closely related to the magnitude of this underestimation. For RMSE (Fig. 4), the corrected DSR shows relatively stable performance when CERES CF underestimation ranges from more than 5% up to more than 30%, with RMSE values increasing only moderately from 28 W/m² to 33.4 W/m². When the CF underestimation further exceeds 30% and reaches 35%, the RMSE rises more sharply, reaching 41.06 W/m². By contrast, the uncorrected DSR (CERES DSR) exhibits a much stronger sensitivity to CF underestimation, with RMSE steadily increasing from 30.57 W/m² at stations with CF underestimation greater than 5% to as high as 46.42 W/m² at greater than 35%. Similarly, the bias (Fig. 4) indicates that the correction substantially reduces systematic deviations. After correction, the bias changes gradually from –0.40 W/m² to 10.13 W/m² across the range of stations with CF underestimation greater than 5% to greater than 35%. In comparison, the uncorrected DSR shows a considerably larger positive bias, increasing from 2.83 W/m² to 33.05 W/m². These results demonstrate that the CF-based correction effectively mitigates both random and systematic errors, particularly under conditions with moderate to large CF underestimation.

305

Figure 4: Validation of DSR against ground-based observations before and after correction for stations with CF differences exceeding various thresholds (Fused\_cf\_Arc – CERES SYN). The x-axis denotes CF difference thresholds (%, e.g., >5%, >10%, ... >35%), while the y-axis represents the RMSE or bias (b). Blue and orange curves correspond to results before and after correction, respectively.

## 4.3 Comparison of DSR before and after correction

Figure 5 presents the spatial distribution of the monthly mean DSR from April to September during the period 2003–2020.

The most pronounced differences between the corrected and uncorrected DSR occur over Greenland, particularly in June, where discrepancies reach up to 30 W/m². Figure 6 further highlights these differences through histograms over Greenland, demonstrating that the correction primarily reduces high DSR values during May, June, July, and August. The correction effects exhibit a distinct seasonal pattern, with more substantial correction during summer. The most notable case is observed in June, where the root-mean-square difference (RMSD) reaches 15.98 W/m², bias is 14.37 W/m², and the maximum per-pixel difference exceeds 35 W/m². In contrast, the correction in September is much less significant, with an RMSD of only 3.83 W/m².

The correction of DSR is influenced not only by the magnitude of CF biases but also by modulation from the solar zenith angle (He et al., 2013, 2015; Kim and Liang, 2010). In May and June, although CF biases are generally larger in May than in June, the correction effect on DSR is more pronounced in June. For example, in northeastern Greenland, pixels with a 39% CF bias in May correspond to a DSR difference of 25 W/m² between the original and corrected values. Whereas in June, pixels with a 32% CF bias exhibit a DSR difference of up to 35 W/m². This discrepancy is primarily attributed to the lower solar zenith angle in June, which enables more solar radiation to reach the surface, thereby amplifying the influence of CF biases on DSR. Consequently, even relatively smaller CF biases can result in more substantial correction effects.

Figure 5: The spatial distribution of the monthly mean DSR for April and September from 2003 to 2020, both before and after correction, as well as the spatial distribution of the differences between the two, and the CF differences (Fused\_cf\_Arc - CERES SYN). All metrics have units of W/m<sup>2</sup>.

Figure 6: Histogram of multi-year average monthly DSR before (red bin) and after (blue bin) correction over Greenland. All RMSD and Bias statistics have units of  $W/m^2$ . The vertical coordinate represents the frequency and the horizontal coordinate represents the DSR value  $(W/m^2)$ .

## 4.4 Validation of estimated all-wave net radiation

The extended model was developed to estimate the monthly mean NR based on the corrected DSR. The LRD was employed to account for the influence of diurnal variations in shortwave radiation on NR across different seasons. As a result, the model does not need to be developed separately for each month; instead, the seasonal effects are implicitly incorporated through the LRD. During validation, the model achieved an R<sup>2</sup> of 0.99, an RMSE of 5.87 W/m<sup>2</sup>, and a bias of 0 W/m<sup>2</sup> on the validation dataset (Fig. 7), demonstrating excellent performance. In addition, the estimated NR was further validated against in situ observations from ground-based stations.


Figure 7: Scatter plot comparing the NR estimated using the extend model with the CERES-SYN NR dataset

Figure 8 presents a comparison of scatterplots between the estimated NR from this study, CERES SYN dataset and ground-based observations. The estimated NR shows clear advantages over CERES NR in terms of overall agreement with ground-based observations. Specifically, the estimated NR achieves an R<sup>2</sup> of 0.77 and an RMSE of 28.9 W/m<sup>2</sup>, compared with 0.61 and 34.88 W/m<sup>2</sup> for CERES NR. Although the bias of the estimated NR is somewhat larger (14.88 vs. 10.08 W/m<sup>2</sup>), the scatter plots indicate that the correction substantially improves the distribution of points along the 1:1 line. In particular, the underestimation present in CERES NR around 50~100 W/m<sup>2</sup> is largely alleviated, leading to a more consistent representation of the observed variability.

Figure 8: Scatterplot comparison between the estimated NR from this study, CERES SYN dataset, and ground-based observations.

We further examined the validation performance under different levels of CF underestimation (CERES SYN relative to Fused cf Arc). This analysis allows us to assess how the CF-based correction behaves under varying degrees of



underestimation. As shown in Fig. 9, validation of the estimated NR against ground-based measurements under different CF difference thresholds further highlights the advantage of using the CF-corrected DSR as the basis for NR calculation. the estimated NR maintains a relatively robust correlation with observations across all CF difference ranges, with R² values decreasing gradually from 0.74 (>5%) to 0.54 (>30%). Although the RMSE (30.21–35.12 W/m²) and bias (16.25–27.41 W/m²) indicate an overall tendency toward overestimation, the scatter plots remain distributed along the 1:1 line, suggesting that the systematic bias is more uniform.

By contrast, the CERES NR shows considerably weaker performance under large CF underestimation. Its R<sup>2</sup> drops rapidly from 0.54 (>5%) to nearly zero (>30%), with scatter plots at CF underestimation greater than 20% exhibiting an almost horizontal distribution, indicating a loss of physical consistency with ground observations. While the RMSE values of CERES NR are comparable to those of the estimated NR, the near-absence of correlation at high CF underestimation suggests that CERES NR fails to capture the variability of surface conditions under such circumstances.

Figure 9: Validation of DSR against ground-based observations before ((g)  $\sim$  (l)) and after ((a)  $\sim$  (f)) correction for stations with CF differences exceeding various thresholds (Fused\_cf\_Arc - CERES SYN).



These results imply that the NR estimated from the CF-corrected DSR provides a more reliable representation of the radiation variability, particularly in terms of correlation structure, even though the magnitude bias remains to be further improved. In other words, the correction ensures that the retrieved NR retains meaningful sensitivity to surface and atmospheric variations, whereas CERES NR tends to collapse under conditions of high CF underestimation. Nevertheless, it should be noted that the performance of the estimated NR is still limited when CF differences are large. Under heavy cloud cover, retrieving NR from satellite observations has long been recognized as an intrinsic challenge. Future efforts may focus on incorporating additional constraints from longwave radiation, atmospheric water vapor, or surface properties to further reduce systematic biases.

Given the pronounced difference of CF over Greenland, we highlight the validation results of NR at selected ground stations in Greenland (Table 4). The results demonstrate that the estimated NR aligns much more closely with ground-based measurements than the original CERES SYN data. Overall, these findings confirm that the proposed extended model is robust. It significantly improves the accuracy of radiation estimates and enhances the reliability and utility of the resulting radiation products.

Table 4: Validation results of the estimated NR and CERES SYN NR against at observation at PROMICE sites in Greenland.

| Site _  | $\mathbb{R}^2$ | RMSE  | Bias   | $\mathbb{R}^2$ | RMSE  | Bias   |
|---------|----------------|-------|--------|----------------|-------|--------|
|         | Estimated NR   |       |        | CERES SYN NR   |       |        |
| KAN_L   | 0.85           | 36.10 | 27.35  | 0.87           | 28.70 | -18.22 |
| KAN_M   | 0.84           | 18.76 | 9.44   | 0.73           | 23.99 | -3.27  |
| KAN_U   | 0.63           | 27.53 | 21.18  | 0.67           | 25.41 | 22.03  |
| KPC_L   | 0.86           | 28.39 | 22.35  | 0.84           | 38.88 | 34.34  |
| $KPC_U$ | 0.64           | 45.90 | 35.61  | 0.79           | 50.70 | 44.55  |
| MIT     | 0.59           | 36.35 | 20.66  | 0.14           | 67.45 | 44.91  |
| NUK_N   | 0.63           | 34.83 | 0.09   | 0.65           | 51.72 | -28.15 |
| NUK_U   | 0.79           | 21.13 | 3.62   | 0.76           | 25.66 | 1.33   |
| QAS_L   | 0.54           | 52.19 | -28.95 | 0.69           | 46.58 | 25.51  |
| SCO_L   | 0.92           | 21.33 | 5.93   | 0.90           | 27.19 | 18.76  |
| $TAS_U$ | 0.79           | 19.16 | -2.98  | 0.54           | 58.05 | 48.39  |
| UPE_L   | 0.91           | 11.36 | -0.05  | 0.84           | 32.49 | -21.88 |
| UPE_E   | 0.85           | 18.39 | 6.65   | 0.74           | 33.84 | -12.97 |





# 4.5 Comparison between the estimated all-wave net radiation and the CERES SYN

As shown in Fig.10, the spatial distribution of the multi-year monthly mean NR over the Arctic from April 2003 to September 2018 is presented for the estimated NR in this study and the CERES SYN. In contrast to the DSR correction results, the differences between the estimated NR and CERES SYN are relatively minor, particularly over Greenland, where the DSR correction had previously shown substantial discrepancies. Moreover, while increased CF typically results in reduced DSR due to enhanced reflection, an opposite pattern is observed in northeastern Greenland during May and July: higher CF is associated with increased NR.

This phenomenon could be attributed to the dual-component nature of NR, which comprises both shortwave and longwave radiation. Clouds reduce the surface energy input by attenuating incoming shortwave solar radiation, while simultaneously enhancing the absorption and re-emission of longwave radiation by the atmosphere, thereby suppressing the loss of longwave radiation from the surface. Notably, in regions such as Greenland, which are characterized by high surface albedo, the contribution of surface-reflected shortwave radiation is relatively large, weakening the net cooling effect of clouds (Miller et al., 2015). As noted by (Kay and L'Ecuyer, 2013), low-level Arctic clouds can exert a net warming effect, as the enhancement of downward longwave radiation often outweighs the reduction in shortwave radiation reaching the surface. The results in this study are consistent with this perspective: in specific regions and months—such as in northeastern Greenland during summer—the enhancement of longwave radiation associated with increased CF is sufficient to offset or even exceed the reduction in shortwave radiation, resulting in a net increase in NR.

Figure 10: The spatial distribution of the monthly mean NR for April and September from 2003 to 2020, estimated NR in this study and the CERES SYN, as well as the spatial distribution of the differences between the two (estimated NR – CERES NR), and the CF differences (Fused\_cf\_Arc – CERES SYN).

## 5 Discussion


## 5.1 Sensitivity of DSR to variations in model input parameters

To assess the relative importance of CF among various input variables, we conducted a sensitivity analysis. Specifically, single input variable was varied individually while all other variables were held constant, and the corresponding DSR responses were evaluated. As shown in Fig. 11, DSR is most sensitive to CF: as CF increases from 5% to 100%, DSR decreases markedly from approximately 315 W/m² to 180 W/m², indicating a strong negative correlation. Cloud top pressure also exerts a non-negligible influence, whereas cloud radius, aerosol optical depth, and relative humidity have relatively weak impacts, suggesting that even large uncertainties in these variables are unlikely to introduce substantial DSR bias.

To investigate the impact of variable covariations on the response structure of DSR, we performed a series of multivariate variation experiments. As illustrated in Fig.12: (a) when only CF and cloud phase are varied, the influence of cloud phase is relatively minor, and the negative DSR response to increasing CF persists clearly; (b) adding cloud base pressure reduces the visibility of its individual impact—previously seen as increasing DSR when exceeding 760 hPa—because the dominant CF effect overrides it; (c) when cloud top pressure is further included, the DSR response exhibits a "rise-then-fall" pattern, indicating that under low CF conditions, other cloud variables can still exert noticeable influence on DSR, but as CF increases, its controlling effect gradually becomes predominant.

We also examined the monthly mean CF over the Arctic (not shown), and found that CF exceeds 40% in most regions. Combined with the analysis in Fig. 12, this suggests that once CF exceeds 40%, DSR consistently decreases with increasing CF. This further confirms that CF is the dominant factor influencing DSR in cloud-rich regions such as the Arctic. Given that CF retrievals from passive remote sensing remain highly uncertain, CF likely represents a major source of current radiative biases. Therefore, even though CF is the only variable available for correction, its dominant influence on DSR makes it sufficient for effectively reducing the bias. The success of our correction further supports the validity of this approach (Figures 2 and 3).

435

Figure 11: Sensitivity of DSR to individual input variables. Each panel shows the DSR response to a single variable, varied across its range while keeping all others fixed. Red lines indicate the mean response.

Figure 12: Sensitivity of DSR to the combined variation of multiple cloud variables. (a) Varying variables include CF and cloud phase. (b) Varying variables include CF, cloud phase, and cloud base pressure. (c) Varying variables include CF, cloud phase, cloud base pressure, and cloud top pressure, and cloud top pressure, and cloud tau.

## 5.2 Advantages of the developed CF perturbation model

In most previous studies, cloud radiative perturbations were typically estimated by constructing kernel functions that relate cloud properties—such as cloud top pressure, cloud optical thickness, geographic location, and month—to radiative responses, or by assuming a linear relationship between cloud changes and radiative perturbations under the assumption of small perturbations (Shell et al., 2008; Soden et al., 2004, 2008). Many studies employed the histogram method, in which the influence of clouds is discretized into multiple cloud regimes. The total radiative response is then obtained by multiplying the occurrence frequency of each cloud regime by the corresponding radiative kernel.

460

465

440 Figure 13 presents example plots of several publicly available cloud radiative kernels, derived from Zelinka et al. (2012) (Fig.13a), Zhang et al. (2021) (Fig.13b), and Zhou et al. (2022) (Fig.13c). All three kernels quantify cloud impacts using a histogram-based approach. We show the cloud radiative kernels for the month of June, corresponding to a cloud top pressure of 500 hPa and a cloud optical thickness of 6. In Fig.13a, the kernel is a function of cloud top pressure, cloud optical thickness, month, latitude, and surface albedo. The kernel shown was selected based on the surface albedo conditions for June. As shown in the figure, even when using the same ISCCP-H simulator dataset, substantial differences can be observed between the cloud radiative kernels of Fig.13a and 13b due to the use of different variable combinations for histogram binning to quantify cloud impacts. This highlights the critical importance of variable selection in kernel construction. In addition, the kernel in Fig.13c, generated using a GCM by Zhou et al. (2022), also exhibits clear differences from the other two, further indicating that the choice of model plays a key role in shaping the structure of cloud radiative kernels.

Figure 13: The spatial distribution of shortwave cloud radiative kernels for June, corresponding to a cloud top pressure of 500 hPa and a cloud optical thickness of 6. Panel (a) is derived from Zelinka et al. (2012), panel (b) from Zhang et al. (2021), and panel (c) from Zhou et al. (2022) The units are W/m²/%.

This study fully acknowledges the high complexity of the atmospheric system, particularly the numerous parameters and nonlinear mechanisms involved in cloud–radiation interactions. To comprehensively characterize the impact of clouds on radiation, we incorporate multiple key factors into the modeling process, including cloud fraction, cloud effective radius, cloud top pressure, cloud optical thickness, and aerosols. We propose a CF perturbation model based on the strong nonlinear fitting capabilities of ANNs. This model effectively captures the response of radiative perturbations to cloud changes under various atmospheric background states and at any location, thereby overcoming the limitations of traditional cloud radiative kernel methods that rely on linear response assumptions.

Our approach avoids the computational burden associated with high-dimensional interpolation in histogram-based kernel methods and exhibits good scalability, enabling direct application to assess the radiative perturbation effects of other variables such as water vapor and aerosols. In addition, the model is entirely trained and constructed using satellite

observational data, without incorporating any information from GCMs or reanalysis products. This helps to eliminate systematic errors often introduced by parameterizations or idealized assumptions in traditional models. In contrast, satellite data offer greater objectivity and more accurately reflect the actual state of the surface–atmosphere system, providing a robust foundation for reliably evaluating the impacts of clouds and other factors on the Earth's radiation balance.

#### **6 Conclusion**







In this study, we developed a CF perturbation model based on an ANN and utilized a more accuracy CF product, Fused\_cf\_Arc, to correct CERES SYN DSR over the Arctic. Building upon this correction, we further established an extended model that enables direct estimation of NR based on the corrected DSR.

By integrating an ANN-based CF perturbation model with the more accurate Fused\_cf\_Arc product, we effectively corrected CERES SYN DSR over the Arctic. The correction substantially improved the agreement with 66 ground-based sites, reducing RMSE from 30.57 to 28.87 W/m² and bias from 2.83 to 0.40 W/m². Importantly, while the accuracy of CERES DSR degraded rapidly with increasing CF underestimation (RMSE rising to 42.44 W/m² and bias to 29.77 W/m² at sites with >30% CF underestimation), the corrected DSR maintained a much lower error level (RMSE 33.48 W/m², bias 10.13 W/m²). The improvements were most pronounced over Greenland, where local corrections reached up to 30 W/m², and seasonal analyses confirmed the enhanced robustness of our method under diverse solar–cloud conditions. These results highlight the critical role of CF accuracy in DSR estimation and demonstrate the effectiveness of our correction in mitigating CF-related biases.

Building upon the corrected DSR, we developed an extended model for directly estimating NR, thereby avoiding the longwave uncertainties inherent in conventional radiative transfer methods. Compared with CERES NR, our NR estimates showed markedly higher consistency with ground measurements, with  $R^2$  increasing from 0.612 (CERES NR) to 0.772 and RMSE decreasing from 34.88 W/m² (CERES NR) to 28.90 W/m². Although the bias rose slightly (from 10.08 W/m² (CERES NR) to 14.88 W/m²), the corrected NR significantly alleviated the underestimation in the 50–100 W/m² range. Importantly, in cases of pronounced CERES CF underestimation (>30%), the CERES NR essentially loses its ability to capture radiation variability ( $R^2 = 0.0182$ ), whereas the estimated NR retains substantial representational skill ( $R^2 = 0.5411$ ). This confirms that NR retrieved from CF-corrected DSR provides a more reliable representation of radiation variability, preserving physical sensitivity to surface and atmospheric processes.

Overall, this study demonstrates a practical and physically consistent framework for improving surface radiation retrievals in the Arctic, where satellite products often suffer from cloud-related uncertainties. By leveraging improved CF information and employing a data-driven approach that bypasses radiative transfer assumptions, the proposed methodology enhances




both the accuracy and interpretability of DSR and NR estimates. These advances are particularly valuable for high-latitude energy budget assessments, climate model evaluation, and monitoring Arctic amplification, and they provide a foundation for extending the method to other regions and for integrating additional radiation corrections in future research.

**Data availability**. All datasets used in this study, as well as the newly generated radiation products, are available from the authors upon request.

**Author contributions**. YZ and TH organized the paper and conducted the related analysis. YZ drew the article graph and prepared the paper. TH and YM conceptualized the paper and revised the whole article. XL modified the paper and provided suggestions for this study. All authors contributed to the discussion of the results and reviewed the paper.

510 Competing interests. The contact author has declared that none of the authors has any competing interests.

Acknowledgements. The authors gratefully acknowledge the providers of the datasets used in this study. CERES data were obtained from NASA Langley Research Center (https://ceres.larc.nasa.gov/). The fused Arctic cloud product is available at Zenodo (https://doi.org/10.5281/zenodo.7624605). MERRA-2 reanalysis data were provided by NASA GMAO (https://gmao.gsfc.nasa.gov/reanalysis/MERRA-2/). The GLASS land surface product can be accessed via GLASS Data Portal (http://www.glass.umd.edu/). Site-based observations were obtained from FLUXNET (https://fluxnet.org/), AmeriFlux (https://ameriflux.lbl.gov/), GEBA (https://www.geba.ethz.ch/), and PROMICE (https://www.promice.org/). The authors sincerely thank all data providers for making these valuable datasets publicly available. The authors gratefully acknowledge the assistance of ChatGPT in polishing the language of this work. The authors are grateful for its contribution in improving the quality of our work.

**Financial support.** This research was supported in part by the National Natural Science Foundation of China (grant No. 42090012), in part by the FengYun Application Pioneering Project and the Innovation Center for FengYun Meteorological Satellite (FY-APP.XC-2023.21).

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
