# Peer review of "Improving Arctic Surface Radiation Estimation Using a Nonlinear Perturbation Model with a Fused Multi-Satellite Cloud Fraction Dataset"

_EGUsphere, 2025_

## Author Comment (AC1)

Dear Reviewer:

We would like to thank you for the time you have put towards our manuscript. All your comments have been considered and responded to carefully, with the resultant revised manuscript being much improved. To facilitate navigation, below, we use a concatenated code of comment number. For instance, C1 means Comment 1. The referees' comments are listed in **black** italics, and our responses in **blue**. New text added to our revised manuscript or heavily revised text in **green**. Please find below our point-by-point replies to your comments.

**General comments:**

This manuscript introduces a new neural network (NN) based approach for correcting estimates of Arctic downwelling shortwave radiation (DSR) and, subsequently, Arctic net surface radiation (NR) from CERES products, cloud properties, and ancillary atmospheric and surface properties. The algorithm improves DSR estimates relative to surface flux observations, especially in cases where CERES underestimates cloud fraction relative to a recently developed cloud product that combines active and passive observations. NR is also improved, though to a lesser degree, likely due to the approach neglecting the varying influences of downwelling longwave radiation (DLR) on NR. There is value to producing more robust DSR estimates that incorporate improved estimates of cloud fraction from active sensors as well as dependences on other atmospheric and surface conditions.

My primary concern with the study is that the NN approach introduces a disconnect between the final DSR and NR estimates and the physics that modulated them. While the multi-variate NN captures nonlinear relationships and includes additional factors that modulate surface radiative fluxes, it masks the precise physical relationships that led to the results. One clear example of this is the fact that NR is estimated from DSR without accounting for cloud or atmospheric influences on longwave radiation. Presumably the NN captures some of the longwave effects through covariances between DSR, DLR, and other regression variables but it cannot make up for the lack of information provided by longwave radiative transfer calculations. The direct physical connection between inputs and simulated fluxes has considerable value for many atmospheric process and climate applications, so it is not clear how this product could be used in those contexts.

Considering both the value of the analysis and the associated concerns, the paper may be suitable for publication after major revisions to better explain which applications the NR product may be address and responding to the following comments.

**DONE.** We thank the reviewer for this important and thoughtful comment regarding the physical interpretability of the NN-based NR estimates. We fully agree that explicit physical connections between radiative fluxes and their controlling processes are essential for many atmospheric process and climate applications. Our approach is not intended to replace physically based radiative transfer calculations, but to provide an empirically constrained estimate of NR under specific observational limitations. Below, we clarify the physical basis and intended scope of our methodology.

First, the relationship between downward shortwave radiation (DSR) and net radiation (NR) is not assumed arbitrarily, but is well established from extensive observations. NR is strongly constrained by incoming shortwave radiation, which represents the dominant energy input to the surface. Numerous empirical studies have demonstrated that a substantial fraction of NR variability can be explained by DSR alone or in combination with a limited number of physically meaningful ancillary variables.

For example, Kjaersgaard et al. (2007) proposed a simple linear formulation to estimate NR directly from DSR using empirically derived coefficients. Kaminsky and Dubayah. (1997) incorporated surface albedo to account for reflected shortwave radiation. Recognizing the role of longwave processes, Iziomon et al. (2000) showed that near-surface air temperature improves NR estimation by constraining net longwave exchange. Subsequent studies extended this framework by introducing variables representing surface and atmospheric states, such as vegetation indices (Wang and Liang, 2009) and relative humidity (Chen et al., 2022; Jiang et al., 2015), to better capture thermodynamic influences on longwave radiation. Collectively, these studies indicate that although longwave radiative processes are physically distinct from shortwave radiation, their contribution is statistically linked to shortwave-driven surface heating and atmospheric thermodynamic conditions. Kjaersgaard et al. (2007) indicate that regression coefficients derived from multi-year observations are less sensitive to interannual anomalies and short-term noise, resulting in more stable and transferable empirical relationships.

Second, the neural network framework is adopted not to obscure physical relationships, but to mitigate the impact of noise, nonlinear interactions, and measurement uncertainties inherent in observational datasets. Simple empirical models rely on fixed functional forms and often struggle to represent the combined effects of surface albedo, vegetation, atmospheric thermodynamic conditions. By contrast, the NN provides a flexible yet data-constrained means of learning statistically robust relationships that are already present in extensive observations, thereby improving the stability and accuracy of NR estimates. Importantly, the NN does not invent radiative processes, but exploits covariances embedded in observations to constrain NR under conditions where explicit longwave radiative transfer information is unavailable.

Finally, we emphasize that the ANN-derived NR product is not intended for process-level attribution of longwave cloud radiative effects or for climate feedback analyses requiring explicit radiative transfer calculations. Its primary utility lies in providing improved NR estimates, particularly in high-latitude regions where biases in shortwave radiation dominate uncertainties in the surface energy budget. The resulting NR fields remain valuable for investigating melt-season surface radiation balance and related processes.

We have revised the manuscript to more clearly state these limitations and to delineate the appropriate applications of the proposed dataset (Lines 406–417):

"Overall, compared with the CERES product, the NR estimated from the improved DSR in this study demonstrates higher accuracy during the warm season, highlighting its enhanced capability to represent the surface radiation balance under melt-season conditions. This dataset is particularly valuable for studies focusing on summer Arctic radiation–related research, such as Oehri et al. (2022)'s investigation of summer energy flux–vegetation interactions and Tjernström et al. (2019)'s analysis of warm-air advection and summer energy budget dynamics. Furthermore, this dataset can be used for cross-validation among multiple products or for analyzing multi-year NR trends during the summer. However, because the NR is derived from DSR, it cannot be produced during the polar night. Unlike CERES, our dataset cannot support all-season climate analyses due to the absence of DSR in winter. Moreover, the ANN-derived NR lacks the physically based interpretability of CERES, which relies on radiative transfer theory and independently retrieved shortwave and longwave fluxes. As such, our product cannot be used for process-level radiation budget studies. Nonetheless, owing to its high accuracy in the warm season, the ANN-based NR provides a useful complementary dataset for warm-season applications, where the improved DSR directly enhances NR accuracy."

Chen, J., He, T., and Liang, S.: Estimation of Daily All-Wave Surface Net Radiation With Multispectral and Multitemporal Observations From GOES-16 ABI, IEEE Trans. Geosci. Remote Sensing, 60, 1–16, https://doi.org/10.1109/TGRS.2022.3140335, 2022.

Iziomon, M. G., Mayer, H., and Matzarakis, A.: Empirical Models for Estimating Net Radiative Flux: A Case Study for Three Mid-Latitude Sites with Orographic Variability, Astrophysics and Space Science, 273, 313–330, https://doi.org/10.1023/A:1002787922933, 2000.

Jiang, B., Zhang, Y., Liang, S., Wohlfahrt, G., Arain, A., Cescatti, A., Georgiadis, T., Jia, K., Kiely, G., Lund, M., Montagnani, L., Magliulo, V., Ortiz, P. S., Oechel, W., Vaccari, F. P., Yao, Y., and Zhang, X.: Empirical estimation of daytime net radiation from shortwave radiation and ancillary information, Agricultural and Forest Meteorology, 211–212, 23–36, https://doi.org/10.1016/j.agrformet.2015.05.003, 2015.

Kaminsky, K. Z. and Dubayah, R.: Estimation of surface net radiation in the boreal forest and northern prairie from shortwave flux measurements, J. Geophys. Res., 102, 29707–29716, https://doi.org/10.1029/97JD02314, 1997.

Kjaersgaard, J. H., Cuenca, R. H., Plauborg, F. L., and Hansen, S.: Long-term comparisons of net radiation calculation schemes, Boundary-Layer Meteorol, 123, 417–431, https://doi.org/10.1007/s10546-006-9151-8, 2007.

Oehri, J., Schaepman-Strub, G., Kim, J.-S., Grysko, R., Kropp, H., Grünberg, I., Zemlianskii, V., Sonnentag, O., Euskirchen, E. S., Reji Chacko, M., Muscari, G., Blanken, P. D., Dean, J. F., Di Sarra, A., Harding, R. J., Sobota, I., Kutzbach, L., Plekhanova, E., Riihelä, A., Boike, J., Miller, N. B., Beringer, J., López-Blanco, E., Stoy, P. C., Sullivan, R. C., Kejna, M., Parmentier, F.-J. W., Gamon, J. A., Mastepanov, M., Wille, C., Jackowicz-Korczynski, M., Karger, D. N., Quinton, W. L., Putkonen, J., Van As, D., Christensen, T. R., Hakuba, M. Z., Stone, R. S., Metzger, S., Vandecrux,

Thorsen, T. J., Kato, S., Loeb, N. G., and Rose, F. G.: Observation-Based Decomposition of Radiative Perturbations and Radiative Kernels, J. Climate, 31, 10039–10058, https://doi.org/10.1175/JCLI-D-18-0045.1, 2018.

Tjernström, M., Shupe, M. D., Brooks, I. M., Achtert, P., Prytherch, J., and Sedlar, J.: Arctic Summer Airmass Transformation, Surface Inversions, and the Surface Energy Budget, J. Climate, 32, 769–789, https://doi.org/10.1175/JCLI-D-18-0216.1, 2019.

Wang, K. and Liang, S.: Estimation of Daytime Net Radiation from Shortwave Radiation Measurements and Meteorological Observations, Journal of Applied Meteorology and Climatology, 48, 634–643, https://doi.org/10.1175/2008JAMC1959.1, 2009.

**Specific Comments:**

**C1**: The introduction is poorly structured and difficult to follow at times. For example, the sentence about limitations in detecting low-level Arctic clouds on Line 55 doesn't seem to directly follow from the sentences preceding it. The paragraph that begins on Line 64 discusses the PRP method needs to be better motivated and connected to the goal of estimating surface radiation.

**DONE.** Thanks for this crucial comment. We have substantially revised the introduction to improve its structure and logical flow. Specifically, the revised text appears in Lines 57–58 and Lines 67-74 of the revised manuscript.

Lines 57-58:

"Despite this advantage, CF still suffers from considerable retrieval uncertainties, which can directly propagate into biases in surface radiation estimates."

Lines 67-74:

"Given that biases in CF can directly propagate into uncertainties in radiation, it is therefore necessary to quantify the specific contributions of CF to the radiation budget. Such quantification provides a pathway for improving radiation estimates when more accurate CF datasets become available. In previous studies, the partial radiative perturbation (PRP) method has been widely used for this purpose because it explicitly separates the radiative influence of individual variables (e.g., CF), thereby offering a direct means to assess how CF errors translate into radiation biases. However, PRP requires repeated radiative transfer calculations and often yields inconsistent feedback estimates across models (Colman, 2003; Soden et al., 2004; Wetherald and Manabe, 1988). These limitations have motivated the development of several related perturbation-based approaches (Table 1)."

**C2**: Line 91: For the general reader, it would be useful to explain what an Angular Distribution Model is and how it works.

**DONE.** Thank you for this helpful suggestion. In the revised manuscript, we have added a concise description of what an ADM is (Lines 100–103).

"The ADM is a core component of the CERES retrieval algorithm, converting radiances observed under specific viewing geometries into broadband TOA fluxes. This conversion is accomplished by applying scene-dependent angular distribution functions that characterize the directional distribution of radiation for different surface and cloud conditions."

**C3**: Equation 7 and the sentences that follow it should appear immediately following its reference in the text on Line 199.

**DONE.** Thank you for pointing this out. We have revised the manuscript accordingly (Lines 210-218).

"…the radiative perturbation $\delta F_{\Delta cf}$ induced by a change $\Delta cf$ in the perturbation variable CF can be directly derived from Eq.7.

$$\delta F_{\Delta cf} = \frac{f_{ANN}(cf + \Delta cf, y_1 ... y_n) - f_{ANN}(cf - \Delta cf, y_1 ... y_n)}{2} + O(\Delta cf^2) \tag{7}$$

where $f_{ANN}$ represents the nonlinear function linking input variables and shortwave radiation. $F_{SYN}$ is the radiation from the CERES SYN, $cf$ is from the CERES SSF CF, $\Delta cf$ represents the difference between the CF in the CERES SSF and the Fused_cf_Arc, non-physical perturbation values are addressed following the rules proposed by Thorsen et al.( 2018), and $O(\Delta cf^2)$ is the truncation error in the ANN model calculation process. Due to the strong capability of ANN in capturing nonlinear dependencies, this residual term is found to be negligibly small and therefore omitted. The corrected shortwave radiation using the improved CF dataset is then written as:

$$F_{corrected} = F_{SYN} + \Delta F_{\Delta cf} \tag{8}$$

The ANN model is trained using the backpropagation (BP) algorithm with 40 neurons in the hidden layer…"

**C4**: Line 221: delete 'budget'

**DONE.** Thank you for this helpful suggestion. We have revised the manuscript accordingly (Line 228).

**C5**: Section 4.4: it is important to note here that deriving NR strictly from DSR, albedo, day length, vegetation cover, temperature, and humidity severely limits the utility of the estimates for process or climate studies. This method essentially assumes that longwave cloud radiative effects can be parameterized from these variables and allows the NN to invent that relationship. Since these relationships are far from unique, physically interpreting the causes of variation in the resulting NR is largely impossible. Even if less accurate based on surface measurements, the independent estimates of DSR and DLR in the CERES product contain more information to enable climate forcing and feedback studies. To provide a concrete example, studies have shown that trends in wintertime surface radiation play a role in ice loss the following year. How can an algorithm that derives NR from DSR provide any insight into the

dark months in the Arctic? The utility of the NR estimates from the ANN approach should be clearly stated as well as the limitations relative to the original CERES product that covers all seasons.

**DONE.** We sincerely thank you for this important and insightful comment. We agree that deriving NR strictly from DSR and other surface/environmental predictors limits the physical interpretability of the resulting NR. This approach constrains the broader range of process-based analyses that can be conducted with products such as CERES. In addition, our method cannot provide NR estimates for the polar night, since the improved DSR serves as the basis for the correction. Consequently, wintertime surface-radiation-related studies cannot be addressed with our NR product.

We would like to clarify that our approach does not aim to parameterize longwave cloud radiative effects or to infer longwave cloud radiative forcing. Instead, it empirically estimates net radiation based on incoming shortwave radiation and a limited set of ancillary variables that represent surface and atmospheric states. The ANN framework exploits statistically robust relationships between DSR, surface heating, atmospheric thermodynamic conditions, and NR that have been well documented in long-term observations, rather than attempting to resolve individual radiative processes.

A substantial body of empirical work has shown that a large fraction of NR variability can be explained by DSR when combined with a small number of physically meaningful predictors (e.g., albedo, temperature, humidity, vegetation)(Chen et al., 2022; Iziomon et al., 2000; Jiang et al., 2015; Kaminsky and Dubayah, 1997; Kjaersgaard et al., 2007; Wang and Liang, 2009). Moreover, long-term observational analyses indicate that these relationships become more stable and transferable when derived from multi-year datasets, as short-term atmospheric variability and interannual anomalies are effectively averaged out(Kjaersgaard et al., 2007). This provides the statistical basis for estimating NR without explicitly resolving longwave cloud processes.

Although restricted to periods with solar illumination, the resulting NR fields remain valuable for investigating melt-season surface radiation balance and related processes. For example, they can support studies focusing on summer Arctic energy exchanges, vegetation–energy interactions (Oehri et al., 2022), or warm-air advection impacts on surface energy budgets (Tjernström et al., 2019). In addition, the dataset can serve as a complementary reference for inter-product comparison or for analyzing multi-year NR variability during the Arctic summer.

Following the reviewer's suggestion, we have revised the manuscript to more explicitly describe both the utility and the limitations of our NR estimates relative to CERES (Lines 406–417)

"Overall, compared with the CERES product, the NR estimated from the improved DSR in this study demonstrates higher accuracy during the warm season, highlighting its enhanced capability to represent the surface radiation balance under melt-season conditions. This

dataset is particularly valuable for studies focusing on summer Arctic radiation–related research, such as Oehri et al. (2022)'s investigation of summer energy flux–vegetation interactions and Tjernström et al. (2019)'s analysis of warm-air advection and summer energy budget dynamics. Furthermore, this dataset can be used for cross-validation among multiple products or for analyzing multi-year NR trends during the summer. However, because the NR is derived from DSR, it cannot be produced during the polar night. Unlike CERES, our dataset cannot support all-season climate analyses due to the absence of DSR in winter. Moreover, the ANN-derived NR lacks the physically based interpretability of CERES, which relies on radiative transfer theory and independently retrieved shortwave and longwave fluxes. As such, our product cannot be used for process-level radiation budget studies. Nonetheless, owing to its high accuracy in the warm season, the ANN-based NR provides a useful complementary dataset for warm-season applications, where the improved DSR directly enhances NR accuracy."

Chen, J., He, T., and Liang, S.: Estimation of Daily All-Wave Surface Net Radiation With Multispectral and Multitemporal Observations From GOES-16 ABI, IEEE Trans. Geosci. Remote Sensing, 60, 1–16, https://doi.org/10.1109/TGRS.2022.3140335, 2022.

Iziomon, M. G., Mayer, H., and Matzarakis, A.: Empirical Models for Estimating Net Radiative Flux: A Case Study for Three Mid-Latitude Sites with Orographic Variability, Astrophysics and Space Science, 273, 313–330, https://doi.org/10.1023/A:1002787922933, 2000.

Jiang, B., Zhang, Y., Liang, S., Wohlfahrt, G., Arain, A., Cescatti, A., Georgiadis, T., Jia, K., Kiely, G., Lund, M., Montagnani, L., Magliulo, V., Ortiz, P. S., Oechel, W., Vaccari, F. P., Yao, Y., and Zhang, X.: Empirical estimation of daytime net radiation from shortwave radiation and ancillary information, Agricultural and Forest Meteorology, 211–212, 23–36, https://doi.org/10.1016/j.agrformet.2015.05.003, 2015.

Kaminsky, K. Z. and Dubayah, R.: Estimation of surface net radiation in the boreal forest and northern prairie from shortwave flux measurements, J. Geophys. Res., 102, 29707–29716, https://doi.org/10.1029/97JD02314, 1997.

Kjaersgaard, J. H., Cuenca, R. H., Plauborg, F. L., and Hansen, S.: Long-term comparisons of net radiation calculation schemes, Boundary-Layer Meteorol, 123, 417–431, https://doi.org/10.1007/s10546-006-9151-8, 2007.

Oehri, J., Schaepman-Strub, G., Kim, J.-S., Grysko, R., Kropp, H., Grünberg, I., Zemlianskii, V., Sonnentag, O., Euskirchen, E. S., Reji Chacko, M., Muscari, G., Blanken, P. D., Dean, J. F., Di Sarra, A., Harding, R. J., Sobota, I., Kutzbach, L., Plekhanova, E., Riihelä, A., Boike, J., Miller, N. B., Beringer, J., López-Blanco, E., Stoy, P. C., Sullivan, R. C., Kejna, M., Parmentier, F.-J. W., Gamon, J. A., Mastepanov, M., Wille, C., Jackowicz-Korczynski, M., Karger, D. N., Quinton, W. L., Putkonen, J., Van As, D., Christensen, T. R., Hakuba, M. Z., Stone, R. S., Metzger, S., Vandecrux,

Tjernström, M., Shupe, M. D., Brooks, I. M., Achtert, P., Prytherch, J., and Sedlar, J.: Arctic Summer Airmass Transformation, Surface Inversions, and the Surface Energy Budget, J. Climate, 32, 769–789, https://doi.org/10.1175/JCLI-D-18-0216.1, 2019.

Wang, K. and Liang, S.: Estimation of Daytime Net Radiation from Shortwave Radiation Measurements and Meteorological Observations, Journal of Applied Meteorology and Climatology, 48, 634–643, https://doi.org/10.1175/2008JAMC1959.1, 2009.

**C6**: Figure 10: following from the previous comment, comparisons of NR are only presented for the daylight months of April – September. Obviously NR cannot be estimated from DSR in winter months but switching between the ANN estimates here back to CERES products in winter would introduce a discontinuity in the physics for analyses spanning the whole year. Again, the utility of the NR estimates introduced here should be clearly stated.

**DONE.** We thank the reviewer for this follow-up comment. We fully agree that combining ANN-derived NR during daylight months with CERES NR during winter would introduce a physical discontinuity and is therefore not recommended for analyses spanning the full annual cycle.

As clarified in Section 4.4, the NR estimates introduced in this study are intended for warm-season applications only and are designed to complement, rather than replace, the original CERES products. Their primary utility lies in improving the accuracy of melt-season net radiation under sunlit conditions, rather than providing a temporally continuous, all-season NR dataset. We have revised the manuscript to explicitly state this limitation and to discourage year-round splicing of ANN-derived NR with CERES products (Lines 406–417).

"Overall, compared with the CERES product, the NR estimated from the improved DSR in this study demonstrates higher accuracy during the warm season, highlighting its enhanced capability to represent the surface radiation balance under melt-season conditions. This dataset is particularly valuable for studies focusing on summer Arctic radiation–related research, such as Oehri et al. (2022)'s investigation of summer energy flux–vegetation interactions and Tjernström et al. (2019)'s analysis of warm-air advection and summer energy budget dynamics. Furthermore, this dataset can be used for cross-validation among multiple products or for analyzing multi-year NR trends during the summer. However, because the NR is derived from DSR, it cannot be produced during the polar night. Unlike CERES, our dataset cannot support all-season climate analyses due to the absence of DSR in winter. Moreover, the ANN-derived NR lacks the physically based interpretability of CERES, which relies on radiative transfer theory and independently retrieved shortwave and longwave fluxes. As such, our product cannot be used for process-level radiation budget studies. Nonetheless, owing to its high accuracy in the warm season, the ANN-based NR provides a useful complementary dataset for warm-season applications, where the improved DSR directly enhances NR accuracy."

Oehri, J., Schaepman-Strub, G., Kim, J.-S., Grysko, R., Kropp, H., Grünberg, I., Zemlianskii, V., Sonnentag, O., Euskirchen, E. S., Reji Chacko, M., Muscari, G., Blanken, P. D., Dean, J. F., Di Sarra, A., Harding, R. J., Sobota, I., Kutzbach, L., Plekhanova, E., Riihelä, A., Boike, J., Miller, N. B., Beringer, J., López-Blanco, E., Stoy, P. C., Sullivan, R. C., Kejna, M., Parmentier, F.-J. W., Gamon, J. A., Mastepanov, M., Wille, C., Jackowicz-Korczynski, M., Karger, D. N., Quinton, W. L., Putkonen, J., Van As, D., Christensen, T. R., Hakuba, M. Z., Stone, R. S., Metzger, S., Vandecrux,

Tjernström, M., Shupe, M. D., Brooks, I. M., Achtert, P., Prytherch, J., and Sedlar, J.: Arctic Summer Airmass Transformation, Surface Inversions, and the Surface Energy Budget, J. Climate, 32, 769–789, https://doi.org/10.1175/JCLI-D-18-0216.1, 2019.

**C7**: Section 5.2 seems out of place in this study. It is not clear how this discussion of radiative kernels relates to the methods introduced in the paper other than to show that radiation dependences on clouds are nonlinear and complicated by atmospheric and surface conditions. No additional insights into cloud effects or feedbacks are gained from this discussion so I would suggest removing or significantly abbreviating it.

**DONE.** Thank you very much for this constructive comment. In response, we have removed this subsection from the revised manuscript.

**C8**: Line 471: 'accuracy' should be 'accurate'

**DONE.** Thank you for pointing this out. We have corrected "accuracy" to "accurate" in the revised manuscript (Line 457).

**C9**: Line 487: The extended model doesn't avoid LW uncertainties, it introduces them but not explicitly accounting for cloud impacts on DLR. This sentence should be modified.

**DONE.** Thank you for this helpful suggestion. Based on your suggestions, we have made the corresponding modifications (Lines 471-472).

"Building upon the corrected DSR, we developed an extended model for directly estimating NR without explicitly retrieving longwave components."

**C10**: Line 500 - 501: In my opinion, applications to climate model evaluation and monitoring Arctic amplification are severely impeded by the lack of explicit consideration of DLR in the NR estimates provided here. It is not clear how one would interpret model biases relative to the output generated from this algorithm or whether the method would adequately capture trends in NR associated with Arctic amplification. It is also not clear how well these methods would extrapolate to other regions where variability may be more closely tied to longwave radiation than shortwave.

**DONE.** We thank the reviewer for this insightful comment. We agree that the lack of explicit DLR consideration imposes limitations on the applicability of our NR estimates. We have revised the corresponding statement in the manuscript (Lines 484–486).

"These advances are particularly valuable for studies focusing on summer Arctic radiation–related research. In addition, the dataset can support cross-validation among multiple products and analyses of multi-year summer radiation trends."